# Tooth Loss and Risk of Lung Cancer among Urban Chinese Adults: A Cohort Study with Meta-Analysis

**DOI:** 10.3390/cancers14102428

**Published:** 2022-05-14

**Authors:** Hyung-Suk Yoon, Xiao-Ou Shu, Yu-Tang Gao, Gong Yang, Hui Cai, Jiajun Shi, Jae Jeong Yang, Nathaniel Rothman, Qing Lan, Wei Zheng, Qiuyin Cai

**Affiliations:** 1Division of Epidemiology, Department of Medicine, Vanderbilt Epidemiology Center, Vanderbilt-Ingram Cancer Center, Vanderbilt University School of Medicine, Nashville, TN 37232, USA; hyung-suk.yoon@vumc.org (H.-S.Y.); xiao-ou.shu@vanderbilt.edu (X.-O.S.); gong.yang@vumc.org (G.Y.); hui.cai@vumc.org (H.C.); jiajun.shi@vumc.org (J.S.); jae.j.yang@vumc.org (J.J.Y.); wei.zheng@vanderbilt.edu (W.Z.); 2Department of Epidemiology, Shanghai Cancer Institute, Renji Hospital, Shanghai Jiaotong University School of Medicine, Shanghai 201112, China; ytgao@vip.sina.com; 3Division of Cancer Epidemiology and Genetics, Occupational and Environmental Epidemiology Branch, National Cancer Institute, Bethesda, MD 20892, USA; rothmann@exchange.nih.gov (N.R.); qingl@mail.nih.gov (Q.L.)

**Keywords:** lung cancer, oral health, tooth loss, smoking, prospective study, meta-analysis

## Abstract

**Simple Summary:**

The risk of lung cancer associated with tooth loss has not been fully addressed, especially for the potential interaction with smoking. This cohort study with meta-analysis first investigated the association between tooth loss and lung cancer risk stratified by smoking status using the Shanghai Men’s and Women’s Health Studies and then summarized epidemiologic findings to date, incorporating the current study and other published studies. Our findings suggest that tooth loss is associated with an increased risk of developing lung cancer, but smoking could modify the association. A meta-analysis of eight epidemiological studies also supports a strong link of tooth loss to lung cancer risk, particularly among current smokers. Improving dental care and oral health may be an effective strategy for lung cancer prevention in addition to smoking cessation.

**Abstract:**

Epidemiological evidence on tooth loss and lung cancer risk remains limited, especially for smoking-specific associations. To investigate the association between tooth loss and lung cancer risk by smoking status, we first analyzed data from the Shanghai Men’s Health Study (*n* = 49,868) and the Shanghai Women’s Health Study (*n* = 44,309). Cox regression models were applied to estimate the hazard ratios (HRs) and 95% confidence intervals (CIs) for lung cancer risk in relation to tooth loss. We also conducted a meta-analysis to summarize epidemiologic findings to date, incorporating results from the current study and six previously published studies. For 7.3 median follow-up years, 973 incident lung cancer cases (613 men and 360 women) were ascertained. After adjustment for major covariates, tooth loss was associated with an increased risk of lung cancer among men (HR [95% CI] for >10 teeth vs. none = 1.59 [1.21–2.11]) but not among women (0.86 [0.50–1.46]). The positive association was stronger among male current smokers (1.75 [1.26–2.45], *p*-interaction by smoking status = 0.04). In a meta-analysis incorporating 4052 lung cancer cases and 248,126 non-cases, tooth loss was associated with a 1.64-fold increased risk of developing lung cancer (relative risk [RR, 95% CI] for the uppermost with the lowest category = 1.64 [1.44–1.86]). The positive association was more evident among current smokers (1.86 [1.41–2.46]), but no significant associations were found among never or former smokers. Our findings suggest that tooth loss may be associated with an increased risk of lung cancer, and the association could be modified by smoking status.

## 1. Introduction

Recently, oral health has gained much attention due to its potential contribution to cancer development and/or prevention [1]. Epidemiologic evidence has indicated that oral health could contribute to the etiology of cancers [2,3,4], including lung cancer [5,6,7,8,9,10]. However, population-based studies to date on tooth loss and lung cancer risk remain elusive, and the potential interaction with smoking remains largely unexplored. Poor oral health, such as gum disease and tooth cavities, can cause tooth loss. Oral pathogens that cause poor oral conditions (e.g., periodontitis and tooth loss) are linked to chronic systemic inflammation, which can promote various types of carcinogenesis by inducing oncogenic mutations, producing tumor-promoting mediators such as cytokines, and stimulating tumor cell proliferation [11]. Of note, some activated cytokines and/or oral bacteria themselves may alter the respiratory epithelium, making respiratory pathogens more susceptible to infection [11,12]; all of which can increase the likelihood of developing lung cancer. Smoking is a significant risk factor for poor oral health [13] and tooth loss [14] and is actively involved in shaping the oral microbiome [15,16]. Smoking toxicants can directly affect oral microbial ecology by promoting antibiotic effects and oxygen deprivation [17]. Given the crucial role of smoking in both lung carcinogenesis and oral health, the association between tooth loss and lung cancer risk may differ by smoking status, but this issue has not been fully addressed. In addition, previous studies evaluating oral health and lung cancer risk were mostly conducted among populations in the United States (USA) [6,7,8,9,10]. Currently, little is known about the association between tooth loss and lung cancer risk in other populations, especially those who have different dental hygiene/care and/or smoking behaviors.

In this study, based on data from the Shanghai Men’s Health Study (SMHS) and Shanghai Women’s Health Study (SWHS), we prospectively investigated the association between tooth loss and lung cancer risk among Chinese populations with a high smoking prevalence in men and a low smoking prevalence in women [18,19] and few standard practices for preventive dental care [3,20]. We further performed a meta-analysis to summarize epidemiologic findings to date, incorporating results from the current study and other published studies.

## 2. Material and Methods

### 2.1. Study Population

The current cohort analyses utilized de-identified resources from the SMHS and SWHS, the population-based, prospective cohort studies conducted in urban Shanghai, China. Detailed information on each cohort profile has been described elsewhere [21,22]. Briefly, 61,480 men and 74,941 women aged 40 to 74 years were recruited between 2002 and 2006 for SMHS and between 1996 and 2000 for SWHS. After obtaining written informed consent, an in-person interview was conducted to collect information on sociodemographic characteristics, medical history, dietary habits, lifestyle factors, and anthropometrics. Active follow-up surveys were repeated every 3–4 years (SMHS) and 2 years (SWHS) to check participants’ health/vital status and obtain further exposure information. Tooth loss information was collected at the second (SMHS) and fourth (SWHS) follow-up surveys by asking about the number of missing teeth: none, 1–5, 6–10, and >10. Of the original study participants, we excluded those with no information on tooth loss (11,496 for SMHS and 30,598 for SWHS) and invalid follow-up time or missing smoking information (116 and 34, respectively). After the exclusion, a total of 49,868 men and 44,309 women remained in the final analytic population. The SMHS and SWHS protocols were reviewed and approved by institutional review boards of the Shanghai Cancer Institute and Vanderbilt University.

### 2.2. Ascertainment of Incident Lung Cancer Cases

Study participants have been followed up for cancer diagnosis and deaths by annual record linkage with databases from the population-based Shanghai Cancer Registry, Shanghai Vital Statistics Registry, and Shanghai Resident Registry (complete rates >99%) and in-person follow-up surveys (follow-up rates >92%). Loss to follow-up occurred due to participants withdrawing their consent, being absent during the study period, or being excluded for other miscellaneous reasons [22]. All possible cancer diagnoses were verified through home visits and a review of medical charts by a panel of oncologists. Lung cancer cases were defined using the International Classification of Disease–Ninth revision (code: 162) and were further subclassified by histologic type: adenocarcinoma, squamous cell carcinoma, small cell carcinoma, or others, according to the morphology code of the International Classification of Diseases for Oncology-second edition. At the end of 2016, we identified 613 and 360 primary incident lung cancer cases with information on tooth loss from SMHS and SWHS, respectively.

### 2.3. Statistical Methods for Cohort Analyses

Given the substantial differences in smoking prevalence across sex, all analyses were carried out separately for men and women. Lung cancer risk factors (i.e., age, smoking status, smoking pack-years, education, household income, alcohol consumption, body mass index [BMI], a history of chronic obstructive pulmonary disease [COPD], and menopausal status for women) were compared between lung cancer cases and cancer-free subjects, using the Student’s *t*-test for continuous variables and the chi-square test for categorical variables. Using Cox proportional hazards regression models, we estimated hazard ratios (HRs) and 95% confidence intervals (CIs) for the development of lung cancer in relation to the number of tooth loss. The ‘none of tooth loss’ category was used as the reference group. Age at tooth loss assessment and age at censoring (i.e., lung cancer diagnosis, death, loss to follow-up, or the latest data linkage/follow-up, whichever came first) were treated as the time scale. A linear trend was tested using the *Wald* test.

Covariates were selected *a priori*, considering potential risk factors identified in our study population, and included in the statistical models sequentially: Model 1 minimally included age at tooth loss assessment (continuous) only; Model 2 further included smoking-related variables such as smoking status (never, former, and current) and pack-years (continuous); and Model 3 (fully-adjusted model) included additional covariates, including education (less than high school, completed high school, and above high school graduation), income (low, medium, and high), alcohol consumption (drinks per day, 1 drink = 14 g of ethanol: none, 0 to ≤2 in men or 0 to ≤1 in women, and >2 in men or >1 in women), BMI (<18.5, 18.5–24.9, 25.0–29.9, or ≥30.0 kg/m^2^), history of COPD (No vs. Yes), and menopausal status (pre vs. post, only for women). The proportion of missing covariates was very low (mostly less than 1%), which were assigned with the median or mode values of non-missing covariates. Stratified analyses were performed to assess the potential effect modification by smoking status. The interaction was tested by the likelihood ratio test, comparing models with/without a cross-product term of tooth loss and smoking status. The joint association between tooth loss and smoking was further assessed using ‘non-current smoking and none of the tooth loss’ as the reference. Sensitivity analyses were conducted, excluding the first 2 years of follow-up, to minimize the possibility of reverse causation and/or the potential influence of preclinical conditions. All analyses were performed using the SAS software, version 9.4 (SAS Institute, Cary, NC, USA). A two-sided *p* less than 0.05 was considered statistically significant.

### 2.4. Meta-Analysis

To identify all available epidemiological studies evaluating the association of tooth loss with lung cancer risk, we systematically searched the PubMed, Embase, and Web of Science databases for articles published in English. The following keywords were used as search terms: “tooth loss” OR “periodontal diseases” OR “oral health” AND “lung cancer” OR “lung carcinoma” OR “lung neoplasm.” If necessary, we performed a manual search from references of selected articles to find further relevant publications. Based on our primary search keywords, we first identified 4982 records in the databases. After initial screening, excluding duplicates or non-original articles, 31 articles were reviewed in detail, but 21 articles were dropped due to lack of relevance to lung cancer (*n* = 17) or tooth loss (*n* = 4). Of 10 articles, we further excluded four [23,24,25,26] because of the different target outcome (lung cancer mortality). Finally, six remaining articles that investigated the association between tooth loss and lung cancer risk were selected for the present meta-analysis [5,6,7,8,9,10]. Three investigators (H.S.Y., J.S., and J.J.Y) worked independently to extract relevant data, which was conducted between October 2021 and December 2021. More details on the article selection process and the selected articles are presented in Appendix A. Including the SMHS and SWHS, we have eight parent studies for the final meta-analysis.

Study-specific risk estimates for the uppermost category of tooth loss vs. none were pooled to compute a summary relative risk (RR) estimate. HRs and odds ratios (ORs) from all included studies were considered as approximate measures of RRs, based on the underlying assumption of the rarity of outcome events [27,28]. Risk estimates among total study participants and by smoking status were combined based on a fixed-effects meta-analysis method if there was no evidence of heterogeneity across studies; otherwise, a random-effects meta-analysis method was applied [29]. The Cochran *Q* test and *I*^2^ statistics were used to assess the between-study heterogeneity: *p*-heterogeneity < 0.10 and/or *I*^2^ > 50% were considered statistically significant heterogeneity [30]. The Egger’s test was conducted to evaluate publication bias [31]. A dose–response meta-analysis was further conducted using the generalized least-squares method [32]. The median value in each category or the midpoint of the upper and lower boundaries was assigned to the corresponding dose of the tooth loss. Restricted cubic splines were used to estimate a potential nonlinear dose–response relationship, with three knots at the 10th, 50th, and 90th percentiles of the tooth loss distribution. All the meta-analyses were performed using STATA version 16.0 (STATA Corp, College Station, TX, USA).

## 3. Results

### 3.1. Cohort Analyses for the SMHS and SWHS

The characteristics of incident lung cancer cases and cancer-free subjects are presented in Table 1. Of the 94,177 participants, 973 primary incident lung cancer cases (613 men and 360 women) were ascertained during the median follow-up time of 7.3 years (median [interquartile range] = 6.6 [5.9–7.7] for SMHS and 7.9 [7.2–8.8] for SWHS). The median time interval between tooth loss assessment and lung cancer diagnosis was 3.9 years (median [interquartile range] = 3.5 [2.0–5.2] for SMHS and 4.5 [2.6–6.2] for SWHS). Regardless of sex, the proportion of current smokers was much higher in lung cancer cases than in non-cases (70.3% vs. 58.8% for men and 5.0% and 1.7% for women; both *p* < 0.001), and cases showed higher smoking pack-years than non-cases (33.6 ± 18.9 vs. 24.1 ± 15.8 pack-years for men and 17.2 ± 11.7 vs. 8.0 ± 10.2 pack-years for women; both *p* < 0.001) among ever smokers. Notably, most women were never smokers (94.2% of lung cancer cases and 98.0% of controls). Male lung cancer cases were more likely to have lower income, lower education attainment, higher alcohol consumption, a higher percentage of BMI less than 25, and a higher proportion of COPD history than cancer-free counterparts (all *p* < 0.05). Meanwhile, female lung cancer cases were more likely to have higher educational attainment and postmenopausal status than non-cases (all *p* < 0.001). When comparing participants’ characteristics across the number of tooth loss (Appendix A), an increased number of tooth loss appeared to be associated with older age, lower education, and higher smoking pack-years in both men and women. Of note, current smoking was significantly associated with higher odds of tooth loss, particularly losing more than 10 teeth, regardless of sex (OR [95% CI] for >10 teeth vs. none = 1.54 [1.37–1.72] for men and 1.73 [1.08–2.78] for women; Appendix A).

Tooth loss was significantly associated with an increased risk of lung cancer only among men (Table 2). When adjusted for age only (Model 1), men showed a significant dose–response association of tooth loss with increased risk of lung cancer: HRs (95% CIs) were 2.23 (1.70–2.93) for >10 teeth, 1.60 (1.17–2.19) for 6–10 teeth, 1.29 (1.00–1.66) for 1–5 teeth, and 1 (ref.) for none; *p*-trend < 0.001. After adjustment for all covariates, including smoking history and other lung cancer risk factors (Model 3), men who lost >10 teeth showed a 59% increased risk of lung cancer (95% CI: 1.21–2.11; *p*-trend <0.001) compared to those without tooth loss. However, no significant association was observed among women (multivariable-adjusted HR = 0.86, 95% CI: 0.50–1.46 for >10 teeth; *p*-trend = 0.84). The sensitivity analysis, excluding the first 2 years of follow-up after oral health assessment, found similar patterns of the observed associations (Appendix A). Cumulative incidence curves by the number of tooth loss are presented in Appendix A.

Results from the stratified analysis by smoking status are presented in Table 3. For male current smokers, losing >5 teeth was significantly associated with an increased risk of lung cancer: multivariable-adjusted HRs (95% CIs) were 1.75 (1.26–2.45) for >10 teeth lost and 1.44 (1.00–2.08) for 6–10 teeth lost; *p*-trend < 0.001. However, non-significant associations were observed among former smokers (HR [95% CI] = 1.63 [0.65–4.07] for >10 teeth lost) or never smokers (HR [95% CI] = 1.05 [0.54–2.05] for >10 teeth lost). We observed a potential interaction between smoking and tooth loss in relation to lung cancer risk (*p*-interaction = 0.04) among men. For women, the number of ever smokers was very small, which resulted in unstable risk estimates: losing >5 teeth appeared to show a positive association with lung cancer risk among current smokers, but the association was not statistically significant (HRs [95% CIs] were 1.51 [0.22–10.4] for >10 teeth lost and 1.78 [0.33–9.69] for 6–10 teeth lost). No association of tooth loss with lung cancer risk was found among female never smokers, which was consistent with men. We further investigated the joint effect of tooth loss and smoking in association with lung cancer risk (Appendix A). Compared to non-current smokers without tooth loss, current smokers who reported having any tooth loss (≥1 tooth) were significantly associated with subsequent development of lung cancer among men (multivariable-adjusted HR [95% CI] = 2.33 [1.50–3.63]), and not significantly associated with lung cancer risk among women (HR [95% CI] = 1.88 [0.97–3.66]), perhaps due to a small number of cases in women.

### 3.2. Results of Meta-Analyses

When combining results from six previous studies and two current cohort analyses, we have a total of 4052 lung cancer cases and 248,126 non-cases for analysis. The overall summary RR for the uppermost category of tooth loss vs. none was 1.64 (95% CI: 1.44–1.86) with no significant heterogeneity (*I*^2^ = 29.1%, *p* = 0.196; Figure 1). The Egger’s test found no evidence for publication bias (*p* = 0.622). Exclusion of the result from a case-control study [5] (i.e., restricted to prospective studies only) had no impact on the overall association (RR = 1.65, 95% CI: 1.44–1.89, *I*^2^ = 38.5%, *p* = 0.135). When the analysis was performed separately for studies conducted in the US and Asia, the RRs were 1.77 (95% CI: 1.50–2.09, *I*^2^ = 0.0%, *p* = 0.513) for the US and 1.44 (95% CI: 1.17–1.77, *I*^2^ = 51.9%, *p* = 0.125) for Asia (Appendix A). When stratified by smoking status (Figure 2), tooth loss was significantly associated with an increased risk of lung cancer among current smokers (RR = 1.86, 95% CI: 1.41–2.46, *I*^2^ = 0.0%, *p* = 0.603), but not among former (RR = 1.32, 95% CI: 0.88–1.98, *I*^2^ = 0.0%, *p* = 0.485) or never smokers (RR = 1.05, 95% CI: 0.79–1.40, *I*^2^ = 0.0%, *p* = 0.895). Publication bias was not detected by the Egger’s test (*p* = 0.631 for current smokers, 0.097 for former smokers, and 0.554 for never smokers). A dose–response meta-analysis (Figure 3) suggested that every five teeth lost was associated with a 9% increased risk of lung cancer among all samples combined (RR = 1.09, 95% CI: 1.07–1.11, *p*-trend < 0.001) and with a 15% increased risk among current smokers (RR = 1.15, 95% CI: 1.09–1.22, *p*-trend < 0.001). However, no significant association was observed among former smokers (RR per every five teeth lost = 1.04, 95% CI; 0.97–1.12, *p*-trend = 0.31) or never smokers (RR per every five teeth lost = 1.03, 95% CI: 0.97–1.08, *p*-trend = 0.39).

## 4. Discussion

Our cohort analyses based on the SMHS and SWHS found that tooth loss was significantly associated with an increased risk of lung cancer among men but not among women. When stratified by smoking status, the positive association remained significant only among current smokers, highlighting a potential effect modification by smoking on the tooth loss and lung cancer association. The results from our meta-analysis also supported a strong link of tooth loss to the development of lung cancer, particularly among active smokers.

In line with our results, previous studies showed a significant association of tooth loss with lung cancer risk. Michaud and colleagues published several shreds of evidence about tooth loss and lung cancer risk: the results from the Health Professionals Follow-up Study showed that fewer teeth were associated with an increased risk of lung cancer (HR for 0–16 vs. 25–32 = 1.70, 95% CI: 1.37–2.21) [6]. However, the significant association disappeared when restricted to never smokers [6,8]. The Atherosclerosis Risk in Communities study reported that people with edentulism had a 2.6-fold increased risk of lung and bronchus cancer compared to those without the condition [9]. In the Southern Community Cohort Study, we observed racial/ethnic differences in the association between tooth loss and lung cancer risk [10]: African–Americans who lost more than 10 teeth showed an increased risk of lung cancer (OR for >10 vs. none = 2.11, 95% CI: 1.03–4.32), but no association was found among European–Americans. Furthermore, a hospital-based study conducted in Japan reported that lung cancer risk increased gradually with the number of teeth lost [5]. Similarly, a previous meta-analysis that adopted the mixed outcome definition (i.e., incidence and mortality) reported a linear dose–response relationship of tooth loss with lung cancer: a 10% increased risk of lung cancer was observed every five increments in tooth loss [33], which was in the same vein of our study findings.

Most previous studies could not fully address subgroup variations by smoking. In our analyses using the SMHS and SWHS, the association of tooth loss with lung cancer risk was modified by smoking status: the primary association was significantly stronger among current smokers but gradually attenuated among former and never smokers (*p*-interaction = 0.04 for men and 0.09 for women). Results from our meta-analyses combining all the existing epidemiological studies up to date, incorporating 4052 lung cancer cases and 248,126 non-cases, also support the smoking-specific associations. Tooth loss and lung cancer risk association substantially differed by smoking status―a stronger association among current smokers, a much-attenuated association among former smokers, and no association among never smokers. Previous studies showed that smoking could play an important role in oral health and be significantly associated with tooth loss [34,35,36]. Cigarette smoking can inhibit the whole-mouth salivary flow rate and cause dry mouth, leading to poor oral health and tooth loss [14,37]. Furthermore, a recent study reported that cigarette smoking could affect the oral microbial ecosystem and change the composition and abundance/prevalence of oral microbiota: the disease-related bacterial taxa such as *Actinomyces graevenitzii* and *T. denticola* were enriched among current smokers [16].

Several other biological mechanisms, particularly those related to inflammation, may also be involved in the associations we observed. Previous studies reported that the number of tooth loss was positively associated with both C-reactive protein and white blood cell counts [38]. People with edentulism were more likely to experience inflammation-related diseases than their dentate counterparts [39]. It is possible that the chronic inflammation associated with tooth loss may contribute, in part, to the development of lung cancer via exacerbating inflammatory and carcinogenic effects of smoking [40]. In addition, poor oral health is strongly associated with the internal production of nitrosamines: tooth loss, a potent indicator of lifetime accumulation of poor oral health, may prompt the production of nitrosamines by nitrate-reducing bacteria [41,42,43], which in turn may contribute to lung carcinogenesis. Furthermore, we observed that certain oral bacteria were associated with the development of lung cancer, suggesting a potential role of the oral microbiome in lung cancer etiology [44,45]. Similarly, a study investigating salivary microbiota among lung cancer patients reported that *Capnocytophaga* and *Veillonella* were higher in the saliva among lung cancer patients [46].

Lung cancer is the leading cause of cancer death in the world, including in China [47]. Cigarette smoking is the dominant cause of lung cancer [18]; thus, smoking cessation is the most important prevention strategy for lung cancer. Many other factors, such as secondhand smoke, radon exposure, indoor and outdoor air pollution, occupational exposure, hormonal factors, certain dietary factors, infections, and chronic lung diseases, are also associated with lung cancer risk [48,49]. In the current study, we found that tooth loss was significantly associated with an increased risk of lung cancer. Notably, standard practices for preventive dental care are relatively lacking in China, especially in rural areas [3,20]. According to the World Health Organization’s Study on Global Ageing and Adult Health (SAGE), the prevalence of edentulism is 8.9% in China [50]. Improving oral health may help reduce the burden of lung cancer in China. Our meta-analysis found that the associations of tooth loss with lung cancer risk are similar in the studies conducted in the US and those in Asia. Improving dental care and oral health could also be an important preventive strategy for lung cancer.

Based on two well-conducted cohorts of Chinese men and women, our study has a large sample size and a wide range of survey data that allowed us to comprehensively investigate the tooth loss associated with lung cancer risk, with adjustment for potential confounders. Via a meta-analysis, furthermore, we systematically summarized the existing evidence to date from prospective studies regarding the association between tooth loss and lung cancer risk overall and by smoking status. However, our study also has several limitations. First, tooth loss was assessed by self-reports, not based on clinical dental examination. Although a previous study reported that tooth loss information collected by trained interviewers was quite accurate [51], measurement errors and misclassifications might affect our findings. However, exposure misclassification in this study is likely to be small and non-differential because tooth loss data was collected before lung cancer diagnosis, which might attenuate the overall associations. Second, oral health data was available for tooth loss only; thus, we could not evaluate other oral diseases such as periodontitis [10,52]. Third, we were unable to account for other important lung cancer risk factors, such as air pollution and occupational exposure (e.g., asbestos), due to a lack of information. Given the significant burden of lung cancer attributable to these factors in China [53,54], future studies with more detailed occupation and environmental exposure information are needed to confirm our findings. Fourth, the current meta-analysis combined HRs and ORs as approximate estimates of RR. There were eight published studies, including seven prospective cohort studies and one case-control study, that met our selection criteria. To increase the sample size, we included all of these eight studies in the meta-analysis. The overall association of tooth loss with lung cancer risk remained the same when the analysis included only seven prospective cohort studies. Finally, the number of lung cancer cases among female smokers was small, prohibiting a stable risk estimation.

## 5. Conclusions

Our findings suggest that tooth loss is associated with an increased risk of developing lung cancer, but the association could be modified by smoking status. A meta-analysis of eight epidemiological studies also supports a strong link of tooth loss to lung cancer risk, particularly among current smokers. In addition to smoking cessation, promoting dental care and improving oral health may be an important lung cancer prevention strategy. Further studies incorporating data on oral microbiome profile, smoking-related biomarkers, and inflammatory biomarkers are warranted to understand better interplays of oral health and smoking in lung cancer etiology.

## Figures and Tables

**Figure 1 cancers-14-02428-f001:**
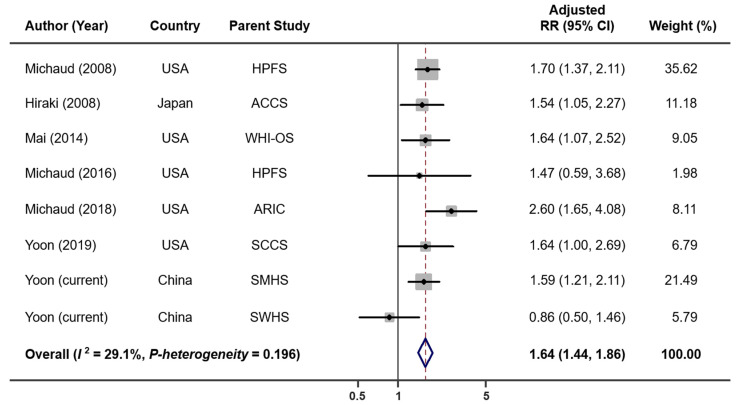
Lung cancer risk associated with tooth loss in each parent cohort. Abbreviations: ACCS—Aichi Cancer Center Study; ARIC—Atherosclerosis Risk in Communities Study; HPFS—Health Professionals Follow-Up Study; RR—relative risk; SCCS—Southern Community Cohort Study; SMHS—Shanghai Men’s Health Study; SWHS—Shanghai Women’s Health Study; WHI-OS—Women’s Health Initiative Observational Study.

**Figure 2 cancers-14-02428-f002:**
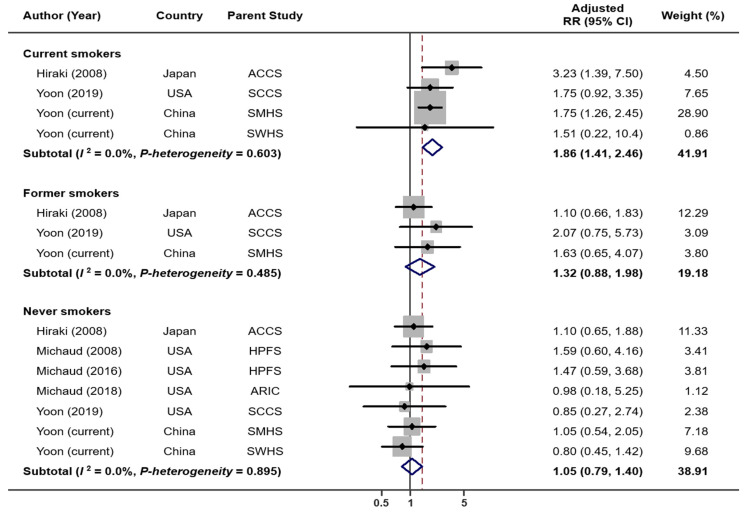
Lung cancer risk associated with tooth loss: by smoking status.

**Figure 3 cancers-14-02428-f003:**
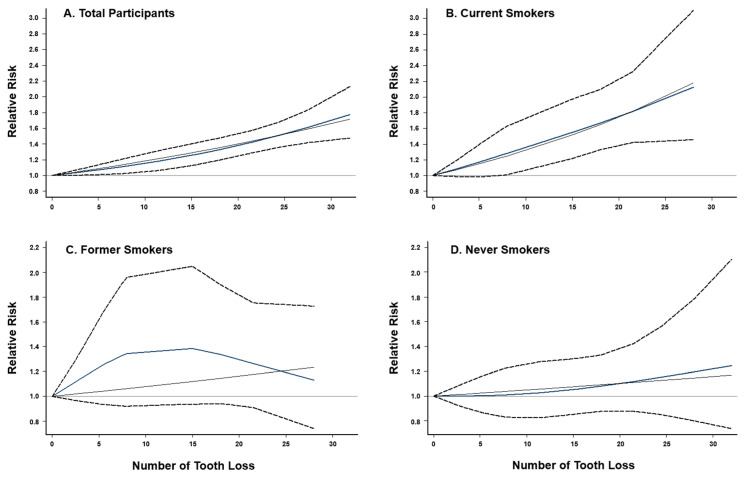
Dose–response relationship between tooth loss and lung cancer risk. (**A**) overall study population (*p* for nonlinearity < 0.001), (**B**) current smokers (*p* for nonlinearity < 0.001), (**C**) former smokers (*p* for nonlinearity = 0.26), and (**D**) never smokers (*p* for nonlinearity = 0.66). Fixed-effects restricted cubic spline models were fitted with three knots at the 10th, 50th, and 90th percentiles. Dash lines represent the pointwise 95% CIs for the fitted nonlinear trend (blue solid lines). Black solid lines represent the linear trend.

**Table 1 cancers-14-02428-t001:** Characteristics of the study population, SMHS and SWHS.

	Men (SMHS)			Women (SWHS)		
Characteristics	LC Cases (*n* = 613)	Non-cases (*n* = 49,255)	*p*-Value ^†^	LC Cases (*n* = 360)	Non-Cases (*n* = 43,949)	*p*-Value ^†^
Age, ^a^ mean (SD)	65.8 (9.6)	60.8 (9.6)	<0.001	64.1 (8.3)	60.8 (8.2)	<0.001
Education						
Less than high school	314 (51.2)	19,740 (40.1)	<0.001	203 (56.4)	25,084 (57.1)	<0.001
Completed high school	176 (28.7)	17,925 (36.4)		84 (23.3)	13,127 (29.9)	
More than high school	123 (20.1)	11,590 (23.5)		73 (20.3)	5738 (13.1)	
Income ^b^						
Low	65 (10.6)	6209 (12.6)	0.03	57 (15.8)	6318 (14.4)	0.72
Medium	504 (82.2)	38,358 (77.9)		270 (75.0)	33,372 (75.9)	
High	44 (7.2)	4688 (9.5)		33 (9.2)	4259 (9.7)	
Smoking Status						
Never	97 (15.8)	15,079 (30.6)	<0.001	339 (94.2)	43,067 (98.0)	<0.001
Former	85 (13.9)	5210 (10.6)		3 (0.8)	118 (0.3)	
Current	431 (70.3)	28,966 (58.8)		18 (5.0)	764 (1.7)	
Pack-years, ^c^ mean (SD)	33.6 (18.9)	24.1 (15.8)	<0.001	17.2 (11.7)	8.0 (10.2)	<0.001
Alcohol consumption ^d^						
None	371 (60.5)	32,904 (66.8)	<0.001	354 (98.3)	43,097 (98.1)	0.91
Low-to-moderate	132 (21.5)	9866 (20.0)		5 (1.4)	677 (1.5)	
Heavy	110 (18.0)	6485 (13.2)		1 (0.3)	175 (0.4)	
BMI (kg/m^2^)						
Under weight, <18.5	36 (5.9)	1996 (4.1)	0.02	3 (0.8)	1373 (3.1)	0.04
Normal, 18.5–24.9	404 (65.9)	30,889 (62.7)		237 (65.8)	27,577 (62.8)	
Overweight, 25.0–29.9	159 (25.9)	15,112 (30.7)		99 (27.5)	12,960 (29.5)	
Obese, ≥30	14 (2.3)	1258 (2.6)		21 (5.8)	2039 (4.6)	
COPD ^e^						
No	496 (80.9)	44,095 (89.5)	<0.001	310 (86.1)	38,441 (87.5)	0.44
Yes	117 (19.1)	5160 (10.5)		50 (13.9)	5508 (12.5)	
Menopausal status						
Pre	-	-	N.A.	144 (40.0)	25,650 (58.4)	<0.001
Post	-	-		216 (60.0)	18,299 (41.6)	
Number of tooth loss						
None	87 (14.2)	13,446 (27.3)	<0.001	94 (26.1)	12,859 (29.3)	0.03
1–5	211 (34.4)	20,430 (41.5)		174 (48.3)	22,565 (51.3)	
6–10	86 (14.0)	5663 (11.5)		75 (20.8)	6844 (15.6)	
>10	229 (37.4)	9716 (19.7)		17 (4.7)	1681 (3.8)	

LC—lung cancer; N—number; SD—standard deviation; BMI—body mass index; COPD—chronic obstructive pulmonary disease; N.A.—not applicable. ^†^ Differences between lung cancer cases and non-cases across characteristics were evaluated by the *t*-test for continuous variables or chi-square test for categorical variables. ^a^ Age at oral health assessment; ^b^ Annual personal income, low, medium, and high, defined as CNY < 4000, ≥4000 to <8000, ≥8000 in the SWHS and CNY < 6000, ≥6000 to <10,000, ≥10,000 in the SMHS, respectively; ^c^ Among current and former smokers; ^d^ Number of total alcoholic drinks (1 drink = 14 g of ethanol) consumed per day was defined as none, low-to-moderate (>0 to ≤2 and >0 to ≤1 drink/day for men and women, respectively), and heavy (>2 and >1 drink/day, respectively); ^e^ Ever diagnosed with emphysema or pulmonary tuberculosis or chronic bronchitis or asthma.

**Table 2 cancers-14-02428-t002:** Association between tooth loss and lung cancer risk, SMHS and SWHS.

				Model 1 ^b^	Model 2 ^c^	Model 3 ^d^
Number of Tooth Loss	Incident Cases, *n*	Person- Years	Incidence Rate ^a^	HR (95% CI)	HR (95% CI)	HR (95% CI)
Men						
None	87	91,068	95.5	1 (ref.)	1 (ref.)	1 (ref.)
1–5	211	137,070	153.9	1.29 (1.00–1.66)	1.19 (0.92–1.54)	1.19 (0.92–1.54)
6–10	86	37,239	230.9	1.60 (1.17–2.19)	1.32 (0.96–1.80)	1.30 (0.95–1.78)
> 10	229	62,592	365.9	2.23 (1.70–2.93)	1.65 (1.25–2.18)	1.59 (1.21–2.11)
*p* for trend				<0.001	<0.001	<0.001
Women						
None	94	102,588	91.6	1 (ref.)	1 (ref.)	1 (ref.)
1–5	174	178,702	97.4	0.86 (0.67–1.11)	0.86 (0.66–1.11)	0.87 (0.67–1.12)
6–10	75	53,650	139.8	1.00 (0.73–1.38)	1.00 (0.72–1.38)	1.01 (0.73–1.40)
>10	17	13,139	129.4	0.86 (0.50–1.46)	0.83 (0.49–1.42)	0.86 (0.50–1.46)
*p* for trend				0.81	0.75	0.84

HR—hazard ratio; 95% CI—95% confidence interval; Ref.—reference. ^a^ Incidence rate per 100,000 person-years. ^b^ Adjusted for age at tooth loss assessment. Age at tooth loss assessment (entry) and age at censoring (exit) were treated as the time scale. ^c^ Adjusted for age at tooth loss assessment, smoking status, and pack-years. Age at tooth loss assessment (entry) and age at censoring (exit) were treated as the time scale. ^d^ Adjusted for age at tooth loss assessment, smoking status, pack-years, education, income, alcohol consumption, BMI, history of COPD, and menopausal status (only for women). Age at tooth loss assessment (entry) and age at censoring (exit) were treated as the time scale.

**Table 3 cancers-14-02428-t003:** Association between tooth loss and lung cancer risk by smoking status, SMHS and SWHS.

			Current				Former				Never	
Number of tooth loss	Incident Cases, *n*	Person- Years	Incidence Rate ^a^	HR (95% CI) ^b^	Incident Cases, *n*	Person- Years	Incidence Rate ^a^	HR (95% CI) ^b^	Incident Cases, *n*	Person- Years	Incidence Rate ^a^	HR (95% CI) ^b^
Men												
None	62	57,533	107.8	1 (ref.)	6	6497	92.4	1 (ref.)	19	27,038	70.3	1 (ref.)
1–5	147	81,974	179.3	1.23 (0.91–1.67)	21	13,014	161.4	1.20 (0.48–3.00)	43	42,082	102.2	1.11 (0.64–1.94)
6–10	62	21,027	294.9	1.44 (1.00–2.08)	13	4449	292.2	1.44 (0.53–3.94)	11	11,763	93.5	0.85 (0.39–1.84)
> 10	160	32,799	487.8	1.75 (1.26–2.45)	45	10,102	445.5	1.63 (0.65–4.07)	24	19,691	121.9	1.05 (0.54–2.05)
*p* for trend				<0.001				0.18				0.92
*p*-interaction												0.04
Women												
None	2	1320	151.5	1 (ref.)	0	159	-	1 (ref.)	92	101,110	91.0	1 (ref.)
1–5	6	2974	201.7	0.88 (0.16–4.70)	1	462	216.5	-	167	175,267	95.3	0.87 (0.67–1.13)
6–10	7	1184	591.2	1.78 (0.33–9.69)	2	181	1105.5	-	66	52,286	126.2	0.95 (0.68–1.33)
>10	3	465	645.2	1.51 (0.22–10.4)	0	92	-	-	14	12,582	111.3	0.80 (0.45–1.42)
*p* for trend				0.37				N.A.				0.55
*p*-interaction												0.09

HR—hazard ratio; 95% CI—95% confidence interval; Ref.—reference; N.A.—not applicable. ^a^ Incidence rate per 100,000 person-years. ^b^ Adjusted for age at tooth loss assessment, pack-years, education, income, alcohol consumption, BMI, history of COPD, and menopausal status (only for women). Age at tooth loss assessment (entry) and age at censoring (exit) were treated as the time scale.

## Data Availability

Data described in the manuscript, codebook, and analytic code will be made available upon request pending application and approval through the Shanghai Women’s and Men’s Health Study online request system (https://swhs-smhs.app.vumc.org/, accessed on 27 November 2021).

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
