# Peer review of "Tooth Loss and Risk of Lung Cancer among Urban Chinese Adults: A Cohort Study with Meta-Analysis"

_cancers, 2022, doi:10.3390/cancers14102428_

Round 1
Reviewer 1 Report
Thank you very much for your work. This is an interesting manuscript and authors may consider the following before it is re-considered for publication. Points in bold are my major concerns.
- Line 48 – Provide references of the cited studies
- Line 75 – See if authors could provide the protocols as a supplementary.
- Line 77 – Provide the reference number of the institution review board approval.
- Line 81 and 82 – Please briefly explain the incomplete follow-up.
- Cox proportional hazards model – One common way of Cox proportional hazards model is to first conduct univariable analysis for each covariate. Only variables with P<0.1 will be considered in the multivariable analysis. Please clarify whether authors analyzed in this way or in other ways.
- Line 118. It is good to have sensitivity analyses. I would be grateful if authors could explain the rationale of “first 2 years of follow-up”, but not other intervals.
- Meta-analysis – I would like to see the detailed search strategy (e.g. in PubMed) as a supplementary.
- Did authors register their systematic review in PROSPERO? Is the meta-analysis done according to PRISMA guidelines?; Please provide the PRISMA checklist.
- Please explain why publication bias test was not conducted.
- Line 140. Relative risks or odds ratios are the study endpoints. However, some of the studies included in this meta-analysis (e.g. Michaud 2008 and the current study by the authors) used hazard ratio as an endpoint, which is time-dependent. It is not an appropriate way to include RR and HR together in a meta-analysis.
- Table 2-3, supplementary Table 3-4 – In addition to person-years, I would like to see the exact incidence (i.e. number of cases per 1000 / 10000 person-years).
- Cumulative incidence curves of lung cancer stratified by number tooth loss and / or smoking status certainly add values.
- Authors should provide related information about the time interval between teeth loss and the diagnosis of lung cancer.
- Some studies in the meta-analysis were conducted in US. As mentioned by the authors in the Discussion, ethnicity is a potential confounder. Subgroup analysis or sensitivity analysis stratified by ethnicity should be conducted to ensure the consistence of the results (e.g. Asian studies). It is especially important as the study objective of this study is to investigate the association of tooth loss and lung cancer in CHINESE adults.
- Supplementary Table 5 – (a) study enrollment—some rows stated the year of study while some stated the study period. Please be consistent. (b) Follow-up duration – state the unit of time (years? Months?). again, some rows stated the median follow-up period while some stated the year in which study ended. Please be consistent. (c) Kindly provide the abbreviations.
- Authors should discuss extensively the implications of this study and any articulations e.g. quit smoking and / or maintain oral health? Is smoking cessation more important than maintaining oral health? How do the findings help healthcare decision-making or policy-making? Any difference between China and US and why? All other meaningful points are appreciated.
- It seems that the cohort study part is the main focus of this study. I suggest the authors modify the study title to “Tooth Loss and Lung Cancer Risk among Urban Chinese Adults: A Cohort Study with Meta-analysis”
- Please improve the resolution of the figures in particular Figure 3.
Author Response
Reviewer #1
Thank you very much for your work. This is an interesting manuscript and authors may consider the following before it is re-considered for publication. Points in bold are my major concerns.
Response: We appreciate Reviewer #1 for taking the time to review our manuscript and provide thoughtful comments and suggestions to improve it. We have revised the manuscript accordingly.
Specific comments:
Comment-1: Line 48 – Provide references of the cited studies
Response-1: We have added the corresponding references.
Line: 51-52 In addition, previous studies evaluating oral health and lung cancer risk were mostly conducted among populations in the United States (US) [6–10].
Comment-2: Line 75 – See if authors could provide the protocols as a supplementary.
Response-2: The study designs of the Shanghai Women’s Health Study and Shanghai Men’s Health study have been published (reference 21 and 22). Detailed study design, baseline and follow-up questionnaires, and cohort characteristics are provided in the study website (https://swhs-smhs.app.vumc.org/), which is publicly available and mentioned in the ‘Data
Availability Statement’.
Comment-3: Line 77 – Provide the reference number of the institution review board approval.
Response-3: We have now provided the approval number and date of the institutional review board to the Institutional Review Board Statement Section.
Line: 354-356 Institutional Review Board Statement: The SWHS (000340; last approval: 07/12/2021) and SMHS (000598, last approval: 12/19/2021) protocols were reviewed and approved by an institutional review board of Vanderbilt University.
Comment-4: Line 81 and 82 – Please briefly explain the incomplete follow-up.
Response-4: We have added an explanation for the incomplete follow-up of the study population in the Material and Methods Section.
Line: 86-88 Loss to follow-up occurred due to participants withdrawing their consent, being absent during the study period, or being excluded for other miscellaneous reasons [22].
Comment-5: Cox proportional hazards model – One common way of Cox proportional hazards model is to first conduct univariable analysis for each covariate. Only variables with P<0.1 will be considered in the multivariable analysis. Please clarify whether authors analyzed in this way or in other ways.
Response-5: Thank you for the comment. We have now clarified the covariate selection.
Line: 109-110 Covariates were selected a priori, considering potential risk factors identified in our study population….
Comment-6: Line 118. It is good to have sensitivity analyses. I would be grateful if authors could explain the rationale of “first 2 years of follow-up”, but not other intervals.
Response-6: The major aim of the sensitivity analyses excluding the first 2 years of follow-up is to confirm that our findings are free from the possibility of reverse causation and/or the potential influence of preclinical conditions. Like many other previous studies, we assume that excluding the first two years of follow-up would be enough and reasonable to minimize the concerns
abovementioned. We performed additional analyses, excluding one or three years of follow-up, yielded almost identical findings. Thus, we present data for excluding first two years of followup.
Line: 124-126 Sensitivity analyses were conducted, excluding the first two years of follow-up, to minimize the possibility of reverse causation and/or the potential influence of preclinical conditions.
Comment-7: Meta-analysis – I would like to see the detailed search strategy (e.g. in PubMed) as a supplementary.
Response-7: We have added detailed information about the article selection process in the revised manuscript.
Line: 130-146 To identify all available epidemiological studies evaluating the associations of tooth loss with lung cancer risk, we systematically searched the PubMed, Embase, and Web of Science databases for articles published in English. The following keywords were used as search terms: “tooth loss” OR “periodontal diseases” OR “oral health” AND “lung cancer” OR “lung carcinoma” OR “lung neoplasm.” If necessary, we performed a manual search from references of selected articles to find further relevant publications. Based on our primary search keywords, we first identified 4,982 records in the databases. After initial screening, excluding duplicates or non-original articles, 31 articles were reviewed in detail, but 21 articles were dropped due to lack of relevance to lung cancer (n=17) or tooth loss (n=4). Of ten articles, we further excluded four [23–26] because of the different target outcome (lung cancer mortality).
Finally, six remaining articles that investigated the association between tooth loss and lung cancer risk were selected for the present meta-analysis [5–10]. Three investigators (H.S.Y., J.S., and J.J.Y) worked independently to extract relevant data, which was conducted between October 2021 and December 2021. More details on the article selection process and the selected articles are presented in Figure S1 and Table S5. Including the SWHS and SMHS, we
have eight parent studies for the final meta-analysis.
Comment-8: Did authors register their systematic review in PROSPERO? Is the meta-analysis done according to PRISMA guidelines?; Please provide the PRISMA checklist.
Response-8: We have not registered our systematic review in the PROSPERO database. Our meta-analysis was conducted in accordance with the Cochrane Handbook for Systematic Reviews of Interventions.
Comment-9: Please explain why publication bias test was not conducted.
Response-9: We have now added the results for publication bias.
Line: 155-156 The Egger’s test was conducted to evaluate publication bias [31].
Line: 222-223 The Egger's test found no evidence for publication bias (p=0.622).
Line: 231-233 Publication bias was not detected by the Egger's test (p=0.631 for current smokers, 0.097 for former smokers, and 0.554 for never smokers).
Comment-10: Line 140. Relative risks or odds ratios are the study endpoints. However, some of the studies included in this meta-analysis (e.g. Michaud 2008 and the current study by the authors) used hazard ratio as an endpoint, which is time-dependent. It is not an appropriate way to include RR and HR together in a meta-analysis.
Response-10: Thank you for the comment. Cohort studies included in our meta-analysis estimated hazard ratios (HRs). In the current meta-analysis, HRs and odds ratios were considered as approximate measures of RR, based on the underlying assumption of the rarity of outcome events. Thus, HRs are interpreted as a simple comparison of two hazards, like an estimate of RR
(i.e., risk ratio), although they are not technically the same. Among the eight studies included in the meta-analysis, seven are prospective cohort studies and one is a case-control study. We performed additional analysis excluding the case-control study (i.e., restricted to prospective studies only), and the overall association remained the same.
Line: 147-150 Study-specific risk estimates for the uppermost category of tooth loss vs. none were pooled to compute a summary relative risk (RR) estimate. HRs and odds ratios (ORs) from all included studies were considered as approximate measures of RRs, based on the underlying assumption of the rarity of outcome events [27,28].
Line: 223-225 Exclusion of the result from a case-control study [5] (i.e., restricted to prospective studies only) had no impact on the overall association (RR=1.65, 95% CI: 1.44-1.89, I2=38.5%, p=0.135).
Comment-11: Table 2-3, supplementary Table 3-4 – In addition to person-years, I would like to see the exact incidence (i.e. number of cases per 1000 / 10000 person-years).
Response-11: As suggested, we have added the exact incidence rate per 100,000 person-years in Table 2-3 and Table S3-S4.
Comment-12: Cumulative incidence curves of lung cancer stratified by number tooth loss and / or smoking status certainly add values.
Response-12: Thank you for the comment. We have added cumulative incidence curves by the number of tooth loss in the SMHS and SWHS. Please see Figure S2.
Line: 197-198 Cumulative incidence curves by the number of tooth loss are presented in Figure S2.
Comment-13: Authors should provide related information about the time interval between teeth loss and the diagnosis of lung cancer.
Response-13: We have provided information on the time interval between tooth loss assessment and the diagnosis of lung cancer.
Line: 168-170 The median time interval between tooth loss assessment and lung cancer diagnosis was 3.9 years (median [interquartile range] = 3.5 [2.0-5.2] for SMHS and 4.5 [2.6-6.2] for SWHS).
Comment-14: Some studies in the meta-analysis were conducted in US. As mentioned by the authors in the Discussion, ethnicity is a potential confounder. Subgroup analysis or sensitivity analysis stratified by ethnicity should be conducted to ensure the consistence of the results (e.g. Asian studies). It is especially important as the study objective of this study is to investigate the association of tooth loss and lung cancer in CHINESE adults.
Response-14: We have conducted subgroup analysis by studies conducted in the USA and in Asia (Supplementary Figure 3). The results are consistent between studies conducted in US and Asia.
Line: 225-228 When the analysis was performed separately for studies conducted in the US and Asia, the RRs were 1.77 (95% CI: 1.50-2.09, I2=0.0%, p=0.513) for the US and 1.44 (95% CI: 1.17-1.77, I2=51.9%, p=0.125) for Asia (Figure S3).
Comment-15: Supplementary Table 5 – (a) study enrollment—some rows stated the year of study while some stated the study period. Please be consistent. (b) Follow-up duration – state the unit of time (years? Months?). again, some rows stated the median follow-up period while some stated the year in which study ended. Please be consistent. (c) Kindly provide the abbreviations.
Response -15: Thank you for the comments. We have updated Supplementary Table 5 as suggested. However, follow-up duration still contains the median follow-up years and the endpoint years because some papers mentioned the year of study endpoint only―in this case, we have described accordingly.
Comment-16: Authors should discuss extensively the implications of this study and any articulations e.g. quit smoking and / or maintain oral health? Is smoking cessation more important than maintaining oral health? How do the findings help healthcare decisionmaking or policy-making? Any difference between China and US and why? All other meaningful points are appreciated.
Response-16: We have discussed the potential implications in the Discussion section.
Line: 296-309 Lung cancer is the leading cause of cancer death in the world, including in China [47]. Cigarette smoking is the dominant cause of lung cancer [18]; thus, smoking cessation is the most important prevention strategy for lung cancer. Many other factors, such as secondhand smoke, radon exposure, indoor and outdoor air pollution, occupational ex-posure, hormonal factors, certain dietary factors, infections, and chronic lung diseases, are also associated with
lung cancer risk [48,49]. In the current study, we found that tooth loss was significantly associated with an increased risk of lung cancer. Notably, standard practices for preventive dental care are relatively lacking in China, especially in rural are-as [3,20]. According to the World Health Organization's Study on Global Ageing and Adult Health Study, the prevalence of edentulism is 8.9% in China [50]. Improving oral health may help reduce the burden of lung cancer in China. Our meta-analysis found that the associations of tooth loss with lung cancer risk are similar in the studies conducted in the US and those in Asia. Optimizing oral health could also be an important preventive strategy for lung cancer.
Line: 333-334 In addition to smoking cessation, promoting dental care may be an important lung cancer prevention strategy.
Comment-17: It seems that the cohort study part is the main focus of this study. I suggest the authors modify the study title to “Tooth Loss and Lung Cancer Risk among Urban Chinese Adults: A Cohort Study with Meta-analysis”
Response-17: Thank you for the comment. The title has been changed to “Tooth Loss and Risk of Lung Cancer among Urban Chinese Adults: A Cohort Study with Meta-Analysis.”
Comment-18: Please improve the resolution of the figures in particular Figure 3.
Response-18: We have prepared the updated Figure 3 with higher resolution and will resubmit it to the editorial office.
Reviewer 2 Report
The aim of the study is to determine whether association exists between tooth loss and lung cancer, and if smoking status modifies this association. The researchers analyzed health data from Shanghai's men and women health studies, and found a positive association between tooth loss and lung cancer among men. They also observed that smoking among men further made this association stronger.
The study considered large number of participants, from two large health studies; conducted meta analyses, and provided detailed results about their observations.
This study is a very important initiative towards underlining the importance of tooth loss association with lung cancer. Further research is necessary to understand the impact of various risk factors like smoking which modifies disease progression. This will help in formulating more effective intervention and treatment strategies in cancer care.
Author Response
Reviewer #2
The aim of the study is to determine whether association exists between tooth loss and lung cancer, and if smoking status modifies this association. The researchers analyzed health data from Shanghai's men and women health studies, and found a positive association between tooth loss and lung cancer among men. They also observed that smoking among men further made this association stronger. The study considered large number of participants, from two large health studies; conducted meta analyses, and provided detailed results about their observations.
This study is a very important initiative towards underlining the importance of tooth loss association with lung cancer. Further research is necessary to understand the impact of various risk factors like smoking which modifies disease progression. This will help in formulating more effective intervention and treatment strategies in cancer care.
Response: We appreciate Reviewer #2 for taking the time to review our manuscript.
Reviewer 3 Report
1. This paper addresses the question of whether tooth loss is a lung cancer
risk factor, either independent or in combination with smoking.
2. This is an interesting and relevant topic, as good dental treatment could
have a preventive effect on lung cancer.
3. The study reports new data from two large Chinese cohorts and also
provides a meta analysis of previous published work,mostly based on Us
populations.
4. It is well controlled for smoking, alcohol,education and female menopausal status, and COPD.
Unfortunately it is not controlled for other important lung carcinogens, both occupational and environmental. This is because of the data bases used. This weakness is not discussed.
5.The methods are excellent and the tables and figures clear. The most convincing evidence for an independent effect is the pack years controlled RR gradient with tooth loss in men (Table 2)
6.References are appropriate apart from lack of reference to other lung carcinogens.
7. error in heading in table 1 (Meman?? should be SWHS) .
Author Response
Reviewer #3
This paper addresses the question of whether tooth loss is a lung cancer risk factor, either independent or in combination with smoking. This is an interesting and relevant topic, as good dental treatment could have a preventive effect on lung cancer. The study reports new data from two large Chinese cohorts and also provides a meta analysis of previous published work, mostly based on Us populations.
Response: We appreciate Reviewer #3 for taking the time to review our manuscript and provide thoughtful comments and suggestions to improve it. We have revised the manuscript accordingly.
Specific comments:
Comment-1: It is well controlled for smoking, alcohol, education and female menopausal status, and COPD. Unfortunately it is not controlled for other important lung carcinogens, both occupational and environmental. This is because of the data bases used. This weakness is not discussed.
Response-1: Thank you for the comment. We have discussed the limitation of controlling other potential risk factors of lung cancer.
Line: 323-327 Third, we were unable to account for other important lung cancer risk factors, such as air pollution and occupational exposure (e.g., asbestos), due to a lack of information. Given the significant burden of lung cancer attributable to these factors in China [53,54], future studies with more detailed occupation and environmental exposure information are needed to confirm our findings.
Comment-2: The methods are excellent and the tables and figures clear. The most convincing evidence for an independent effect is the pack years controlled RR gradient with tooth loss in men (Table 2)
Response-2: We appreciate your comment.
Comment-3: References are appropriate apart from lack of reference to other lung carcinogens.
Response-3: We have discussed other risk factors for lung cancer.
Line: 298-301 Many other factors, such as secondhand smoke, radon exposure, indoor and outdoor air pollution, occupational exposure, hormonal factors, certain dietary factors, infections, and chronic lung diseases, are also associated with lung cancer risk [48,49].
Comment-4: error in heading in table 1 (Meman?? should be SWHS).
Response-4: Thank you for pointing this out. We have revised the heading in Table 1.
Reviewer 4 Report
Thank you for submitting your work to Cancers Journal, evaluating the link between the tooth loss and risk of lung cancer among urban Chinese adult and perfumed a meta-analysis.
- There is no an enough information in the introduction related to a link of tooth loss and risk of lung cancer (your topic), but you linked the attributing factors such as smoking and bad oral health to a risk of lung cancer. Please elaborate the pathogenesis between tooth loss and risk of lung cancer supplemented with citations.
- Page 3, line 130: Could you identify the period of search?
- Supplement Figure 1. Article selection process for the systematic review and meta-analysis. This is need to be referred, as a Flowchart with citation.
- Following your above statement. I suggest to modify your title to "Tooth Loss and Risk of Lung Cancer among Urban Chinese Adult: Systematic Review and Meta-analysis.
Author Response
Reviewer #4
Thank you for submitting your work to Cancers Journal, evaluating the link between the tooth loss and risk of lung cancer among urban Chinese adult and perfumed a meta-analysis.
Response: We appreciate Reviewer #4 for taking the time to review our manuscript and provide thoughtful comments and suggestions to improve it. We have revised the manuscript accordingly.
Specific comments:
Comment-1: There is not an enough information in the introduction related to a link of tooth loss and risk of lung cancer (your topic), but you linked the attributing factors such as smoking and bad oral health to a risk of lung cancer. Please elaborate the pathogenesis between tooth loss and risk of lung cancer supplemented with citations.
Response-1: We have provided additional rationale for studying tooth loss with lung cancer risk in the Introduction Section.
Line: 38-46 Poor oral health, such as gum disease and tooth cavities can cause tooth loss. Oral pathogens that cause poor oral conditions (e.g., periodontitis and tooth loss) are linked to chronic systemic inflammation, which can promote various types of carcinogenesis by inducing oncogenic mutations, producing tumor-promoting mediators such as cytokines, and stimulating tumor cell proliferation [11]. Of note, some activated cytokines and/or oral bacteria themselves may alter the respiratory epithelium, making respiratory pathogens more susceptible to infection [11,12]; all of which can increase the likelihood of developing lung cancer.
Comment-2: Page 3, line 130: Could you identify the period of search?
Response-2: We have identified the period of search in the manuscript.
Line: 142-143 Three investigators (H.S.Y., J.S., and J.J.Y) worked independently to extract relevant data, which was conducted between October 2021 and December 2021.
Comment-3: Supplement Figure 1. Article selection process for the systematic review and metaanalysis. This is need to be referred, as a Flowchart with citation.
Response-3: We have provided the detailed information of article selection process in the revised manuscript.
Line: 130-146 To identify all available epidemiological studies evaluating the associations of tooth loss with lung cancer risk, we systematically searched the PubMed, Embase, and Web of Science databases for articles published in English. The following keywords were used as search terms: “tooth loss” OR “periodontal diseases” OR “oral health” AND “lung cancer” OR “lung carcinoma” OR “lung neoplasm.” If necessary, we performed a manual search from references of selected articles to find further relevant publications. Based on our primary search keywords, we first identified 4,982 records in the databases. After initial screening, excluding duplicates or non-original articles, 31 articles were reviewed in detail, but 21 articles were dropped due to lack of relevance to lung cancer (n=17) or tooth loss (n=4). Of ten articles, we further excluded four [23–26] because of the different target outcome (lung cancer mortality).
Finally, six remaining articles that investigated the association between tooth loss and lung cancer risk were selected for the present meta-analysis [5–10]. Three investigators (H.S.Y., J.S., and J.J.Y) worked independently to extract relevant data, which was conducted between October 2021 and December 2021. More details on the article selection process and the selected articles are presented in Figure S1 and Table S5. Including the SWHS and SMHS, we
have eight parent studies for the final meta-analysis.
Comment-4: Following your above statement. I suggest to modify your title to "Tooth Loss and Risk of Lung Cancer among Urban Chinese Adult: Systematic Review and Meta-analysis.
Response-4: Thank you for the comment. The title has been changed to “Tooth Loss and Risk of Lung Cancer among Urban Chinese Adults: A Cohort Study with Meta-Analysis”
Round 2
Reviewer 1 Report
Thank you very much for your revised manuscript. Authors have addressed all of my concerns in a satisfactory way. Just one minor comment: authors should also briefly discuss the combined application of HR/OR/RR as one of the study limitations. I consider it is acceptable to be published in Cancers after authors address this minor concern.
Author Response
Thank you very much for your revised manuscript. Authors have addressed all of my concerns in a satisfactory way. I consider it is acceptable to be published in Cancers after authors address this minor concern.
Response: We truly appreciate Reviewer #1’s comments.
Specific comments:
Comment-1: Authors should also briefly discuss the combined application of HR/OR/RR as one of the study limitations.
Response-1: Thank you for the comment. We have now addressed this issue as one of the study limitations.
Line: 327-330
Fourth, the current meta-analysis combined HRs and ORs as approximate estimates of RR. There were eight published studies, including seven prospective cohort studies and one case-control study, met our selection criteria. To increase the sample size, we included all these eight studies in the meta-analysis. The overall association of tooth loss with lung cancer risk remained the same when the analysis included only seven prospective cohort studies.
Reviewer 4 Report
The authors have addressed my comments and suggestions. Hence, I accept the manuscript in its present form for publication. Congrats.
